# The Effect of Low-intensity Aerobic Training Combined with Blood Flow Restriction on Maximal Strength, Muscle Mass, and Cycling Performance in a Cyclist with Knee Displacement

**DOI:** 10.3390/ijerph19052993

**Published:** 2022-03-04

**Authors:** Fabiano Aparecido Pinheiro, Flávio Oliveira Pires, Bent R. Rønnestad, Felipe Hardt, Miguel Soares Conceição, Manoel E. Lixandrão, Ricardo Berton, Valmor Tricoli

**Affiliations:** 1School of Physical Education and Sport, University of São Paulo, São Paulo 05508-030, Brazil; hardt@usp.br (F.H.); conceicao.miguel0106@gmail.com (M.S.C.); manelixef@gmail.com (M.E.L.); ricardoberton88@gmail.com (R.B.); vtricoli@usp.br (V.T.); 2Exercise Psychophysiology Research Group, School of Arts, Science and Humanities, University of São Paulo, São Paulo 03828-000, Brazil; piresfo@usp.br; 3Section for Health and Exercise Physiology, Inland University of Applied Sciences, 2624 Lillehammer, Norway; bent.ronnestad@inn.no

**Keywords:** vascular occlusion, knee osteoarthritis, muscle hypertrophy, cycling time-trial

## Abstract

Low-intensity aerobic training combined with blood flow restriction (LI + BFR) has resulted in increases in aerobic and neuromuscular capacities in untrained individuals. This strategy may help cyclists incapable of training with high intensity bouts or during a rehabilitation program. However, there is a lack of evidence about the use of LI + BFR in injured trained cyclists. Thus, we investigated the effects of LI + BFR on aerobic capacity, maximal isometric strength, cross-sectional area of vastus lateralis (CSA_VL_), time to exhaustion test (TTE), and 20 km cycling time-trial performance (TT20 km) in a male cyclist with knee osteoarthritis (OA). After a 4-week control period, a 9-week (2 days/week) intervention period started. Pre- and post-intervention TT20 km, peak oxygen consumption (VO_2peak_), power output of the 1st and 2nd ventilatory thresholds (1st W_VT_ and 2nd W_VT_), maximum power output (W_max_), TTE, muscle strength and CSA_VL_ of both legs were measured. Training intensity was fixed at 30% of W_max_ while the duration was progressively increased from 12 min to 24 min. There was a reduction in time to complete TT20 km (−1%) with increases in TT20 km mean power output (3.9%), VO_2peak_ (11.4%), 2nd W_VT_ (8.3%), W_max_ (3.8%), TTE (15.5%), right and left legs maximal strength (1.3% and 8.5%, respectively) and CSA_VL_ (3.3% and 3.7%, respectively). There was no alteration in 1st W_VT_. Based on the results, we suggest that LI + BFR may be a promising training strategy to improve the performance of knee-injured cyclists with knee OA.

## 1. Introduction

Knee overuse injuries are one of the most common problems affecting professional and competitive cyclists (23%) [1]. These injuries can either impair cycling performance or prevent the athlete from taking part in training sessions and thus reduce training stimulus and racing performance [1]. Therefore, training strategies capable of mitigating knee injuries and their pain-related symptoms are of interest.

The use of low-intensity aerobic training (i.e., 30% of maximal power output (W_max_); LI) combined with blood flow restriction (BFR) may be an effective strategy to stimulate physiological adaptations [2]. It has been observed that combined LI + BFR increases aerobic capacity (i.e., peak oxygen consumption (VO_2peak_) in physically active individuals [3] and cyclists [4,5], and muscle cross sectional area (CSA), muscle strength and performance in open loop exercise in physically active individuals [3,6]. One may argue that cyclists with knee injuries could benefit from LI + BFR training programs, as these athletes are frequently hindered from performing high-load training sessions such as high intensity interval training (HIIT) and/or maximum strength training. In this regard, the LI + BFR could provide a training stimulus to keep knee-injured cyclists at a training status comparable to that before the injury without increasing the risks of an injury aggravation. However, no study has investigated the benefits of the LI + BFR in such condition. Therefore, the aim of the present case report was to investigate the effect of LI + BFR on aerobic capacity, maximal isometric strength, vastus lateralis CSA (CSA_VL_), time to exhaustion, and 20 km time-trial performance (TT20 km) in an amateur male cyclist with knee osteoarthritis (OA) acquired after knee displacement and anterior cruciate ligament reconstruction.

## 2. Materials and Methods

### 2.1. Subject

The subject was a male cyclist and in 2005, at nearly 30 years old, suffered a motorcycle accident that resulted in total left knee displacement with immediately reconstructive surgery. He started systematic cycling training one year after the surgery, participating in several road cycling and time-trial events. Over the years, the subject’s left knee developed OA what caused a limitation of that joint to support high-intensity training loads. At the time of the present study (2018), the subject was characterized as a trained cyclist according to classification criteria from Jeukendrup et al. [7]. Additional subject characteristics are shown in Table 1. He was informed about the experimental procedures, risks, and benefits before providing a written informed consent. This study conformed to the Declaration of Helsinki and was approved by the Institutional Research Ethics Committee (nº 3.235.872).

### 2.2. Experimental Overview

During the first visit, the subject was subjected to anthropometrical measurements and familiarization sessions with maximal voluntary isometric contraction (MVIC) and the TT20 km tests. Thereafter, in the second visit, he performed a TT20 km as a “control test” before initiating a “control period” lasting for the next four weeks. This period was adopted to monitor the cyclist’s performance and his habitual cycling training. After the control period, pre-intervention tests were performed in the following order: visit 3—TT20 km test; visit 4—maximal incremental test (MIT) to determine the VO_2peak_, first and second ventilatory thresholds (VT_1_ and VT_2_, respectively), and W_max_; visit 5—time to exhaustion test (TTE); visit 6—MVIC test and visit 7—measurements of the CSA_VL_ of both legs. The post-intervention tests followed the same order, with the exception of the CSA_VL_ measurements, which were conducted 72 h after the last training session. Figure 1 shows the experimental overview. The subject was instructed to refrain from intense exercise on the day before the tests. An interval of 48–72 h between all testing sessions (visits from 1 to 7) was given in pre- and post-intervention. The subject was also oriented to refrain from caffeinated foods and beverages (i.e., coffee, chocolate, etc.) for 24 h before the tests. All tests were conducted at the same time of day in similar environmental conditions with controlled temperature (21 °C) and monitored humidity (71.4 ± 3.3%).

### 2.3. Cycling Performance Tests and Physiological Measurements

All cycling tests were performed on a Speed bicycle (Soul, Ventana, Itajaí, SC, Brazil) coupled with a cycle-simulator (CompuTrainer^TM^ RacerMate^®^ 8000, USA) calibrated before each test according to manufacturer’s instructions. Bicycle’s adjustments were made according to cyclist’s preference (saddle height, distance of saddle tip top bar, and handlebar position) using their own cycling shoes and pedals. All adjustments were reproduced in following tests.

Before starting the TT20 km, a 10-min warm-up at 150 W (9-min in free pedal cadence and gear, and 1-min in a controlled model with fixed gear between 80–100 rpm) was performed with a 10 s countdown at the end of warm-up to start the trial. Subject was free to choose pacing strategy, pedal cadence, gears, and drink water. Information about distance covered was available and verbal encouragement was given every 2 km with an additional verbal encouragement in the 19th km.

Before starting the MIT, baseline measurements were done for three minutes followed by a 5-min warm-up at 150 W and 80–100 rpm. Thereafter, increments of 25 W·min^−1^ were applied with a pedal cadence between 80–100 rpm until exhaustion, which was defined as the inability to maintain a pedal cadence ≥80 rpm even with verbal encouragements. The subject wore a mask (Hans Rudolph, Kansas, USA) connected to an open-system gas analyser for breath-by-breath measures (Quark CPET, Cosmed, Italy) and VO_2_ was continuously recorded. The VO_2_ data were averaged to 10 s intervals and the VO_2peak_ was determined as the average of the three highest VO_2_ values in the last 60 s of the test [8]. Additionally, the power output corresponding to VT_1_ and VT_2_ (WVT_1_ and WVT_2_, respectively) was also identified [9]. The W_max_ was calculated as the power output of the last stage with the correction of the time of permanence in the incomplete stage when necessary.

The TTE began with a 5-min warm-up at 150 W with immediately increment of the power output corresponding to VO_2peak_ (WVO_2peak_) and 80–100 rpm until exhaustion (inability to keep ≥80 rpm even with verbal encouragements). The time was recorded in seconds and was used as a performance criterion. Importantly, the post-intervention test was performed with the same absolute power output from pre-intervention test.

### 2.4. Maximal Voluntary Isometric Contraction and Muscle Cross-Sectional Area

The MVIC test was performed on a dynamometer (Biodex System 4, Biomedical Systems, Newark, CA, USA). The subject seated on the device’s chair with his hips at 90° and knees at 60° from horizontal. The estimated knee joint center of rotation was aligned with the dynamometer center of rotation. The subject’s chest and hips were fixed with Velcro straps to avoid accessory movements. After, a specific warm-up (50%, 60%, and 70% 5-s MVIC trials separated by 60 s rest intervals) was performed. Then, the subject was oriented to perform two 5-s attempts with 60 s intervals aiming to reach peak torque with the knee joint at 60° from the horizontal. Knee extensor peak torque was used for the statistical analysis. Seventy-two hours after the MVIC test and without any cycling activity, CSA_VL_ of both legs was measured using ultrasound equipment (SonoAce R3, Samsung-Medison, Gangwon, Korea) following the same procedures described in Lixandrão et al. [10]. Briefly, after 10 min in the supine position an experienced researcher found the midpoint between the greater trochanter and the inferior border of the lateral epicondyle of the femur. This point was transversely marked every 2 cm with semipermanent ink from the thigh internal-to-lateral direction to orient the probe positioning (B-mode ultrasound using a 7.5 MHz linear-array). Bony landmarks and probe placement were recorded and documented to ensure reproducible probe placement in the post-intervention assessment. After, the legs were restrained with Velcro straps to avoid hip internal or external rotation during the assessments. Then, sequential ultrasound images were acquired transversally following the ink marks in the skin following an internal-to-lateral direction. CSA_VL_ was determined using a digital software (Image J—National A. P. Hayashi et al. Institutes of Health, Bethesda, MD, USA). The coefficient of variation of the measurement was 1.69%.

### 2.5. Determination of Training Occlusion Pressure

After 10 min in the supine position, a vascular doppler (DV-600; Marted, São Paulo, Brazil) probe was placed over the right tibial artery of the subject to determine total occlusion blood pressure (mmHg). A cuff (Missouri, São Paulo, SP, Brazil) with dimensions of 17.5-cm width by 92-cm length attached on the proximal portion of the thigh (inguinal fold region) was inflated up to the point at which the auscultatory pulse of the tibial artery was interrupted. Training occlusion pressure was set at 80% of the total occlusion pressure and remained constant throughout the intervention period.

### 2.6. Training Program

Before each training session a warm-up (5 min at 150 W, 80–100 rpm) was performed and then a cuff belt was applied on the subject’s thighs with ~134 mmHg of pressure. Eighteen sessions were performed over nine weeks (2×/week). Training was divided into three phases, each lasting three weeks. Every three weeks an MIT was performed to adjust training power output (30% W_max_). In the first phase each training session was composed of four sets of 3 min. Second-phase sessions were composed of 2 × 3 sets of 3 min while in the third phase 2 × 4 sets of 3 min were applied. Each repetition was defined by 2 min cycling plus 1 min passive but with BFR. The cuffs were deflated for five minutes only during the rest interval between sets of exercise in the second and third phases. In each set, the subject exercised for two minutes (30% W_max_, 80–100 rpm) and remained for one minute in passive rest. Habitual cycling training volume (hours/week) was monitored through Strava App (Strava, San Francisco, CA, USA) and intensity was assessed using sessions rating of perceived exertion [11]. In addition, habitual training intensity was classified into low, moderate, and high [12].

### 2.7. Statistical Analysis

Data are reported as means ± standard deviations, when appropriate. Additionally, delta change (%) was calculated between pre- to post-intervention.

## 3. Results

After the 4-week control period we observed a stabilization in TT20 km performance (time = 0.3%, W_mean_ = −1.4%, and W_mean_ (W·BW^−1^) = −3.0%) (Table 1). On the other hand, when comparing pre- to post-intervention a higher mean power output and shorter time to complete the TT20 km were found (Table 1). Accordingly, increases in VO_2peak_, absolute and relative W_max_, WVT_2_, and TTE were also found (Table 1). There was no alteration in WVT_1_ (Table 1). As shown in (Figure 2A), the left leg (injured knee) had a greater increase in MVIC with a discrete increase in the right leg, while both legs achieved similar increases in CSA_VL_ from pre- to post-intervention (Figure 2B). In relation to habitual cycling training, there were no substantial differences in volume or intensity during the study (Table 2).

## 4. Discussion

The novel finding of the present study was that LI + BFR improves TT20 km performance, aerobic capacity, maximum strength, and CSA_VL_ in a trained cyclist with knee OA.

A cycling endurance training program is the main strategy to improve cycling performance [13]. Additionally, other strategies such as HIIT and maximal strength training associated with cycling endurance training may help to improve performance [13,14,15]. The use of both strategies is undoubtedly important to stimulate physiological capacities and increase strength and muscle mass. However, these strategies may not be suitable in individuals with knee OA. Thus, the LI + BFR protocol used in the present study may be an attractive alternative.

The use of LI + BFR has showed significant improvements in aerobic capacity, isometric strength, quadriceps CSA and performance in physically active, non-injured individuals [3,6,16]. In recreational cyclists there was only an increase in VO_2max_ (4.5%) with no improvement in TT15 km performance when BFR was applied between sprint intervals [5]. Our results are in agreement with Abe et al. [3] and de Oliveira et al. [6] which also used BFR during the exercise execution. The results suggest that perhaps exercising with BFR for long periods increases the metabolic stress inducing adaptations such as angiogenesis and mitochondrial biogenesis [2,3,5,6] contributing to aerobic performance. Furthermore, adaptations in maximal strength and muscle mass have been associated with enhanced cycling performance in well-trained cyclists after 12-weeks of maximal strength training [14]. The present study indicates that the LI + BFR was also efficient in enhancing MVIC (mainly in the injury leg) and CSA_VL_ (Figure 2) without increasing the risk of injury aggravation due to the low training intensity that was used (30% W_max_). These adaptations may have contributed to the improvements in cycling performance and its determining variables (Table 1 and Figure 2) observed in our study.

It is important to highlight that although the 9-week intervention period has provided a small increase in the total cycling training volume (~0.4 h·week^−1^), the intensity used in our study (30% W_max_) was below the minimum threshold necessary to stimulate increases in the aerobic capacity of healthy individuals [17]. Furthermore, the LI alone would not provide an adequate stimulus to induce muscle mass and strength enhancement.

### Limitations

Although considerable effort was done to confirm the cycling performance stabilization using a control period, it is important to point out that this period was shorter than the intervention period (4-week versus 9-week, respectively). However, according to the subject’s cycling history, we believe that 4 weeks were sufficient to confirm the stabilization. In addition, there was a lack of physiological measures (i.e., heart rate) during habitual cycling training that, in conjunction with session RPE, would help us to improve monitoring the time spent in different training intensity zones. Another limitation that should be mentioned is the lack of reliable data for the MVIC measurement. Although the subject had been well familiarized with the test, we interpreted this result with caution.

## 5. Conclusions and Practical Recommendations

This case study results suggest that LI + BFR may be a promising training strategy to improve TT20 km performance, aerobic capacity, maximum strength, and muscle CSA in knee-injured cyclists with knee OA. The results corroborate previous LI + BFR training studies conducted with non-injured participants showing improvements in aerobic performance predictor variables, muscle strength and CSA, and time to complete a TT20 km.

In addition, our findings indicate that LI + BFR associated with habitual cycling training may be an interesting protocol to stimulate physiological adaptations and improve performance in trained cyclists. Therefore, LI + BFR training can be used as a rehabilitation strategy and as an integral part of training programs in a broad range of cyclists. However, future studies with large samples are needed to confirm our results and to better understand the adaptations generated by LI + BFR in trained cyclists.

## Figures and Tables

**Figure 1 ijerph-19-02993-f001:**
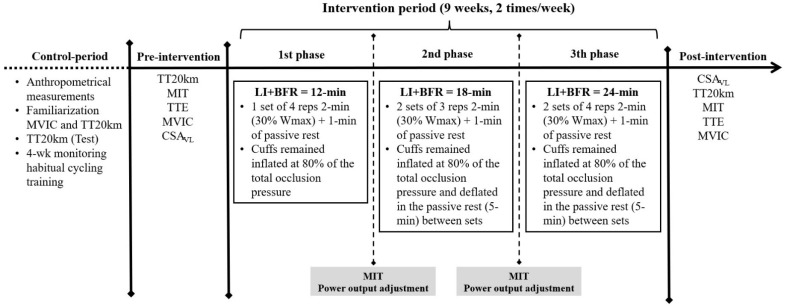
Experimental overview.

**Figure 2 ijerph-19-02993-f002:**
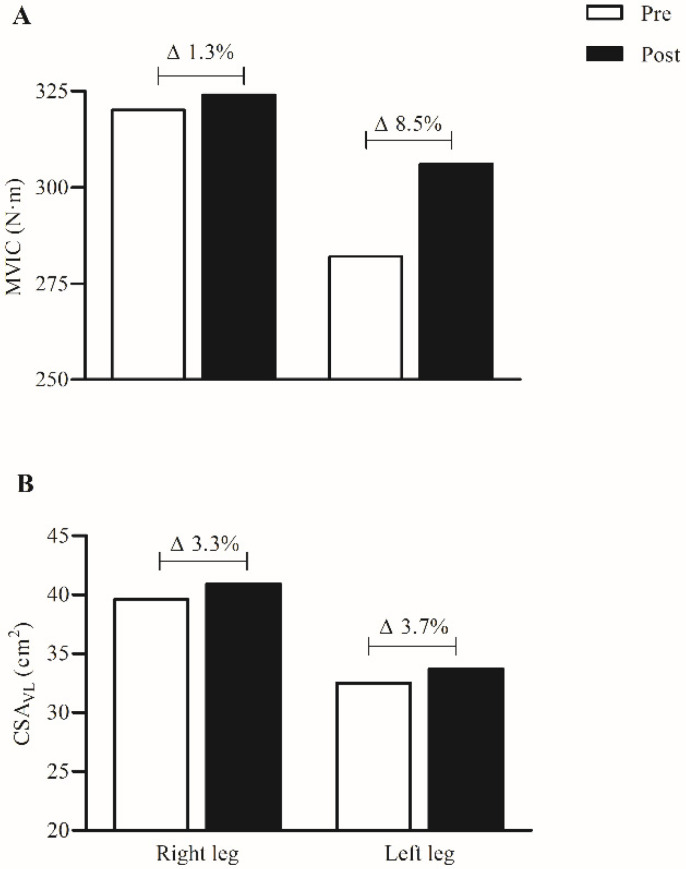
(**A**) maximal voluntary isometric contraction (MVIC) pre (white bars) versus post-intervention (black bars) of the right and left legs; (**B**) vastus lateralis cross-sectional area pre- (white bars) versus post-intervention (black bars) of the right and left legs.

**Table 1 ijerph-19-02993-t001:** Male cyclist anthropometrical, physiological and performance characteristics.

Variables	Control Period	Pre- Intervention	Post- Intervention	Difference (%) (Pre vs. Post)
BW (kg)	81	81.3	79.3	−2.5
Height (cm)	173	173	173	0.0
TT20 km (min)	32.2	32.3	32.0	−1.0
TT20 km W_mean_ (W)	265.5	261.7	272.0	3.9
TT20 km W_mean_ (W·BW^−1^)	3.3	3.2	4.4	37.5
VO_2peak_ (mL.kg^−1^·min^−1^)	n.a	49.2	54.8	11.4
W_max_ (W)	n.a	363.3	377.1	3.8
W_max_ (W_max_·BW^−1^)	n.a	4.5	4.8	6.7
WVT_1_ (W)	n.a	225	225	0.0
WVT_2_ (W)	n.a	300	325	8.3
TTE (s)	n.a	238	275	15.5

BW = body weight; TT20 km = time trial 20 km; TT20 km W_mean_ = mean power output during TT20 km; TT20 km W_mean_ (W·BW^−1^) = mean power output relative to body weight during TT20 km; VO_2peak_ = peak oxygen consumption; W_max_ = maximal minute power output during the incremental test; W_max_ (W_max_·BW^−1^) = maximal minute power output relative to body weight during the incremental test; WVT_1_ = power output at the first ventilatory threshold; WVT_2_ = power output at the second ventilatory threshold; TTE = time to exhaustion at W_max_; n.a = not applicable.

**Table 2 ijerph-19-02993-t002:** Volume (duration) and intensity (RPE) of the habitual cycling training performed during the control period (four weeks) and the intervention period (nine weeks).

Volume/Intensity	Control Period	Intervention Period
Duration (h·week^−1^)	6.2 ± 0.7	6.6 ± 1.3
%RPE_session_ < 4	22.8 ± 3.4	23.4 ± 5.5
%RPE_session_ > 4 and <7	38.8 ± 3.5	37.3 ± 2.1
%RPE_session_ > 7	38.4 ± 5.8	39.4 ± 5.3

%RPE_session_ = percentage of the total duration performed at a specific rating of perceived exertion (<4, >4 and <7 and >7).

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
