# Peer review of "The Effect of Low-intensity Aerobic Training Combined with Blood Flow Restriction on Maximal Strength, Muscle Mass, and Cycling Performance in a Cyclist with Knee Displacement"

_ijerph, 2022, doi:10.3390/ijerph19052993_

Round 1
Reviewer 1 Report
As a case report, the present paper focused on the effect of low-intensity aerobic training combined with blood flow restriction rehibilation strategy on a professional cyclist with knee displacement, and the result showed positive promotion on athlete's exercise performance according to relative indexes. The topic is advanced in rehabiilation study aera, and the work was represented approriately. Here are some minor questions and sugguestions:
(1) On the training protocol, what is the support materials? Why not to improve the number and frequency of training groups using the 18-session training strategy for 9 weeks.
(2) LI+BFR is the focus of discussion in this paper, why is it not included in the keywords? Is "Muscle Hypertrophy" an important indicator of experimental results?
(3) The authors indicate that no studies have investigated the benefits of LI+BFR. Is there any literature supporting the low intensity range of 30% Wmax set in this paper? Why increase the duration from 12 minutes to 24 minutes? Is it scientific to coordinate blood flow restriction?
(4) The author described in detail the process, equipment, test indexes and experimental procedures of the exercise intervention. However, the differences between athletes' cycling performance and cycling habit observed in the first four weeks of the control experiment are not strong after LI+BFR intervention. Can this part be integrated?
(5) The indexes of lateral leg without knee OA could be a perfect negtive control.
Reviewer 2 Report
The Authors have submitted an interesting case report. The manuscript was well developed.
Nevertheless the one consists of some shortcomings, which should be supplemented / corrected before publishing.
Remarks and questions:
Title: Currently it sounds as if the conclusion. I move to change the title in compliance with the ʺInstructions for Authorsʺ (https://www.mdpi.com/journal/ijerph/instructions).
Abstract: The abstract wasn`t elaborated in compliance with the ʺInstructions for Authorsʺ. There is a lack of ʺBackgroundʺ.
Introduction: I would suggest writting the aim of the study in more clear way (the lines 49-54).
Materials and Methods:
Section 2.2. Experimental Overview: There is no information about the lags (time intervals) between ʺvisit 1ʺ , ʺvisit 2ʺ and so on. The main rule is that the procedure should be described with sufficient details to allow others to replicate the research. I must add that the manuscript lacks a diagram showing the time course of the research. It would be clear information for a reader.
Section 2.6. Training Program: Please let me know what the authors mean under the term ʺsetʺ in phrases ʺ4 sets of 3 minʺ, ʺ2x 3 sets of 3 minʺ, ʺ2x4 sets of 3 minʺ?
Discussion
The line 221: Please justify the quoted literature from 1984. It seems that the publication is very old and the data contained in it may be out of date.
Reviewer 3 Report
General comments
This manuscript aims at investigating the effect of low-intensity aerobic training combined with blood flow restriction on peak oxygen consumption, maximal power output, maximal voluntary isometric contraction, vastus lateralis muscle cross sectional area and 20-km cycling time-trial performance in an amateur cyclist with knee osteoarthritis acquired after knee displacement and anterior cruciate ligament reconstruction. Taking into account the limitations regarding case studies, authors manage to fulfil sufficiently their aim.
Minor comments
(line 64) … et al. [7]. Additional…
(l175) … Figure 1 (panel A)…
(l209÷12) please, split;
(l217) … et al. [3] and…
(l236÷40) please, split.
Author Response
Please see the attachment

This manuscript is a resubmission of an earlier submission. The following is a list of the peer review reports and author responses from that submission.